## Research Article

Latin America; trauma; prenatal; postnatal; brief treatment

**Corresponding author:**
Laura Miller-Graff;
Email: lmiller8@nd.edu

# Assessing potential added benefits of trauma-focused content to a guided low-intensity psychoeducational intervention for perinatal women: A propensity score-matched analysis of a nonrandomized trial

Laura Miller-Graff[1] [ID], Jessica Carney[2], Elsa Padilla Cancino[3] and Liliana Yataco Romero[3]

[1]Department of Psychology, Kroc Institute for Peace Studies, Keough School of Global Affairs, University of Notre Dame, Notre Dame, IN, USA; [2]Department of Psychology, University of Notre Dame, Notre Dame, IN, USA and [3]Instituto de Pastoral de la Familia, Lima, Peru

## Abstract

Brief, low-intensity interventions may hold untapped promise for bolstering maternal health in low-resource contexts. The current study used propensity score matching (PSM) to evaluate uptake and differential effectiveness of two low-intensity digital perinatal health (PH) support programs in Lima, Peru. Pregnant women ($N = 251$) were assigned to one of two conditions (PH vs. trauma-focused PH [TF-PH]) and received weekly psychoeducational content via WhatsApp from a lay paraprofessional for 5 weeks. Conditions were not randomly assigned; PSM was used to improve causal inference of the condition. Women were interviewed before participation (T1), immediately following treatment (T2) and at 3 (T3) and 12 months postpartum (T4). Intimate partner violence had strong negative effects on women's mental health, multisystem resilience and parenting, and single mothers reported higher levels of depression and posttraumatic stress symptoms than did partnered women. Intervention uptake was high, with 77% of women participating in all sessions. There were no significant differences between treatment groups over time, but effect sizes indicated a slight advantage of the TF-PH condition in addressing depression symptoms ($d_r = -0.29$) and multisystem resilience ($d_r = 0.39$). Study findings suggest that brief interventions may be well-received and that trauma-focused supports may also confer additional benefits for addressing depression and resilience.

## Impact statement

The current study was a pilot trial of a digital intervention (i.e., via WhatsApp) implemented by local lay paraprofessionals to support perinatal mental health in an under-resourced community in Lima, Peru. Across both treatment conditions, intervention uptake was high. Findings suggest that past-year intimate partner violence and being a single mother were risk factors for participants' mental health and well-being. Although there were no significant differences between treatment groups over time, evaluation of effect sizes indicated that the treatment condition with added trauma-focused content in addition to perinatal health content yielded better results in addressing participants' depression symptoms and multisystem resilience from baseline to 3 months postpartum. This study demonstrates that brief, digital interventions administered by lay paraprofessionals are feasible in this setting and may promote positive adjustment for maternal mental health and parenting in the perinatal period. This is particularly promising given the low availability of maternal mental health care in many global contexts and may support increased access to mental health supports during this key developmental juncture.

## Introduction

Globally, an estimated 27% of women experience common perinatal mental health disorders (Abdul Aziz et al., 2025). Perinatal mental health difficulties increase risk for adverse physical and mental health outcomes for both mothers and children during pregnancy, labor, delivery and the postpartum period (Cook et al., 2018; O'Dea et al., 2023; Runkle et al., 2023). Yet, maternal mental health is often neglected in large-scale public health initiatives (Atif et al., 2015). In contexts with limited mental health supports, low-intensity interventions delivered by lay paraprofessionals may hold promise as a frontline approach to addressing perinatal mental

health at scale. Low-intensity therapeutic interventions are usually brief, leverage technological supports and promote broad dissemination of high-quality, interpretable information that enables self-guided learning (Bennett-Levy et al., 2010). They are designed to dramatically increase availability and enhance more equitable access to evidence-based care (Bennett-Levy et al., 2010). The current study aimed to evaluate the uptake and potential differential effects on maternal perinatal depression, post-traumatic stress (PTSS), multisystem resilience and parenting of two clinician-guided low-intensity interventions delivered to pregnant women living in the San Juan de Lurigancho district of Lima, Peru. Both programs were delivered electronically (via text message through WhatsApp) with one-on-one support from a trained lay paraprofessional; one program focused on perinatal health alone (PH) and one program integrated the PH information with additional trauma-focused content (TF-PH).

### Social context, adversity and perinatal mental health

Perinatal mental health difficulties do not arise in a vacuum; rather, they are influenced by multisystemic processes of risk and protection. One such process is intimate partner violence (IPV; psychological, physical and sexual violence by current or former partner). IPV prevalence is between 38.7 and 45.1% for Peruvian women of reproductive age, with an estimated prevalence of 21% during pregnancy (Gomez-Beloz et al., 2009; Perales et al., 2009; Burgos-Muñoz et al., 2021). Violence against children is also common. The Instituto de Nacional de Estadística e Informática (INEI, 2016) estimates that ~80% of Peruvian children have experienced physical or sexual violence. These high rates of interpersonal violence co-occur with other experiences of community and sociopolitical violence. For example, San Juan de Lurigancho, the setting of the current study, was declared to be in a "state of emergency" due to high rates of gang violence and crime by the Peruvian government in 2023 (Wilder, 2024).

Exposure to violence and adversity has significant implications for perinatal mental health. Peruvian women who experience perinatal IPV are at significantly higher risk for postpartum depression; one large study found that women exposed to prenatal IPV had a 5.9- to 10.8-fold increased risk for moderate and severe depression (respectively) compared to those who had not (Gomez-Beloz et al., 2009). Histories of child abuse are also associated with a greater risk of experiencing IPV and are independently associated with risk for antepartum depression in Peru (Barrios et al., 2015). Additionally, experiencing child abuse and IPV have both been found to predict prenatal PTSS severity in Peru (Sanchez et al., 2017; Carney et al., 2025).

There is also growing evidence that maternal exposure to violence poses substantial risks to parenting. Peruvian women exposed to IPV report lower levels of parental warmth, less energy for parenting and spending less time doing caregiving activities (Kohrt et al., 2015). Peruvian caregivers with higher rates of exposure to traumatic events have also been found to engage in harsher parenting behaviors (Scheid et al., 2021). In other contexts, the presence of IPV and maltreatment has been associated with low parenting morale (Malta et al., 2012).

### Social ecological resilience and perinatal mental health

Despite the negative impacts of violence exposure on mothers' mental health and parenting, many women also demonstrate high levels of resilience. In Ungar's social ecological theory of resilience,

resilience refers to the ways individuals navigate to and negotiate for resources that promote positive adaptation; in this model, resilience is conceptualized as a range of individual, relational and cultural/contextual resources across social ecologies (Ungar & Liebenberg, 2011). For pregnant women who have experienced childhood adversity, it has been found that relational resources are protective against depression (Howell et al., 2020). In Peru, research on resilience in pregnancy has found that women's adaptability, persistence and coping are associated with lower levels of postpartum depression in the context of household dysfunction (Carroll et al., 2021). Peruvian women's prenatal multisystem resilience has also been associated with lower prenatal PTSS (Carney et al., 2025). Thus, it is critical that supports for pregnant women not only address challenges that women may face in the aftermath of violence, but also facilitate resource access and recognize strengths.

### Providing mental health care in resource-constrained contexts

There are substantial barriers to the quality, availability and accessibility of mental health supports for pregnant women in low-resource, high-violence contexts. Across global contexts, low-intensity interventions have been identified as a promising approach to help address the gap between the need for mental health supports and the availability of care, which cannot be addressed with traditional, individualized and face-to-face care alone (Bockting et al., 2016). San Juan de Lurigancho, which has a population of over one million, is served by just four community mental health centers (Gale, 2023). Infrastructure challenges disrupt implementation of evidence-based care in this context, including knowledge gaps and overwhelming service demand (Arriola-Vigo et al., 2019). Given the low availability of care in this context, guided low-intensity interventions delivered by lay paraprofessionals may support broadening access to needed supports, yet there is little research in this context to inform their use.

Low-intensity interventions include treatments that involve lower usage of a specialist therapist compared to traditional individual therapy models, as well as briefer treatments, those delivered via telemedicine and interventions delivered by paraprofessionals (Sijbrandij et al., 2020). With regards to perinatal mental health, there is evidence to suggest that relatively low-intensity psychoeducation on mental health may have incremental benefits for perinatal mental health (Park et al., 2020; standardized mean difference (SMD) = −0.30) and parenting (Yuen et al., 2022). Of note, supporting women through psychoeducation on health and childbirth more broadly may also confer benefits for perinatal mental health (Ngai et al., 2009). Digital mental health interventions also demonstrate strong evidence of effectiveness, although the magnitude of effect varies depending upon intervention intensity, participant engagement and nature of the control condition (Firth et al., 2017; Fu et al., 2020). Together with the fact that the effect sizes of universal prevention programs tend to be small as a function of including a broad range of the population relative to the target outcomes (Tanner-Smith et al., 2018), it would be expected that a guided, digitally implemented perinatal psychoeducational intervention would have small effects. Importantly, small effect sizes should not necessarily be interpreted as poor evidence of effectiveness; rather, they should be considered in light of the intensity of the intervention and its costs, as well as its contribution to the overall system of care in the region. That is, an intervention with a small but discernibly positive effect may be well worth the effort from a systems perspective if it provides low-cost "frontline" care that reduces overall clinical burden in the population, allowing

for the allocation of more intensive services to those with more acute needs. Moreover, in settings where access to care is scarce, even a low-intensity intervention with small effects may result in substantial public health impact.

A key question for these interventions, however, is how best to support treatment engagement and adherence. Self-guided interventions often result in failed uptake (i.e., users never access the content) or in relatively minimal adherence across modules (e.g., Owen et al., 2015; Hanano et al., 2022; Singla, 2024). As such, low-intensity interventions must implement strategies to facilitate *intervention uptake.* One such strategy that has been shown to support adherence and minimize attrition is human support by clinicians or lay paraprofessionals (Werntz et al., 2023). Clinician support of low-intensity interventions refers to brief, targeted supports that help participants navigate self-guided content (e.g., sending reminders, clarifying technical problems, ensuring understanding and providing referrals).

Trauma-informed approaches may be especially important for such programs in settings with high levels of violence. Trauma-informed care focuses on educational strategies that promote trauma awareness and support for both providers and clients (Berliner and Kolko, 2016). Unfortunately, few trauma-informed approaches to low-intensity intervention have included robust evaluations of effectiveness (Berliner and Kolko, 2016). Further, effectiveness data on the added value of trauma-specific content to broader psychoeducational supports is sorely needed.

### The current study

The current study aimed to compare the effectiveness of two guided low-intensity interventions for pregnant Peruvian women living in a resource-constrained context by comparing women who received psychoeducation on PH with and without the integration of trauma-specific content. However, it became evident in the first in-person meeting following the pandemic that well-meaning study staff had assigned some women to the TF-PH condition if they felt that the women would particularly benefit from this content based on their responses to the baseline surveys. Analyses of group differences at baseline suggested several differences between groups at baseline (see Electronic Supplement 1). As such, the study was considered nonrandomized, and the original analytic plan (i.e., for this to be evaluated as a randomized controlled trial) was adjusted. Given that factors confounded with failed randomization were both known and measured (i.e., women's responses to the trauma-related questions on the baseline survey), propensity score matching (PSM) was used as a way of evaluating program effects under imperfect (but real-life) conditions to adjust for baseline confounders (Caliendo and Kopeinig, 2008). Of note, a systematic review found that studies that use PSM produce consistent treatment effect estimates compared to randomized clinical trials (Kitsios et al., 2015), indicating that this is a reliable way to assess treatment effects in real-world conditions when randomization is not achieved.

The first study aim was to describe uptake of the two intervention programs via engagement with WhatsApp-delivered treatment content. The second aim was to examine the differential effectiveness of the two programs on pregnant women's depression symptoms, PTSS, multisystem resilience and parenting confidence, controlling for exposure to IPV, adverse childhood experiences (ACEs) and single motherhood. Maternal PTSS, depression, multisystem resilience and parenting confidence were selected as the targeted treatment outcomes given evidence from other contexts

that brief interventions for violence-exposed pregnant women can have significant, enduring effects on these factors (e.g., Miller-Graff et al., 2022; Howell et al., 2024; Martinez-Torteya et al., 2025).

## Methods

### Setting

Data were collected in the San Juan de Lurigancho district of Lima, Peru. San Juan de Lurigancho is an urban community with over one million inhabitants (World Bank, 2017). San Juan de Lurigancho's residents are exposed to high levels of poverty, crime and interpersonal violence (Scheid et al., 2021; United Nations High Commission on Refugees, 2017). Residents perceive San Juan de Lurigancho to be highly dangerous (Flores et al., 2021) and experience a dearth of resources due to poverty (INEI, 2020). Many families in San Juan de Lurigancho live in self-constructed homes that are not connected to formal resources like potable water and electricity (World Bank, 2017; Perleche-Ugás et al., 2022). There are only two state hospitals in San Juan de Lurigancho, prompting difficulties in accessing healthcare, such as prenatal or labor and delivery care (Perleche-Ugás et al., 2022). Health centers in San Juan de Lurigancho are grouped by micronetworks. The area of the current study is served by the Ganimedes Micronetwork, which includes three health centers (Ganimedes, Huáscar II and Huáscar XV) and two health posts (Ayacucho and Medalla Milagrosa).

### Participants

Participants were eligible if they were currently pregnant, Spanish speaking, at least 14 years of age and living in San Juan de Lurigancho. The minimum age was determined through preparatory work for the study, including focus groups, conversations with local health care providers and project staff. These conversations reflected concerns that were consistent with population-level data, indicating the high prevalence of adolescent pregnancy (30% of sexually active adolescents; Caira-Chuquineyra et al., 2024). Participants ranged in age from 16 to 44 years ($M$ = 28.02, standard deviation [SD] = 6.72). Most participants were married or living with a partner (81.6%); 10.6% were single, and 4.7% were separated from their partners. Average monthly household income was low ($M$ = 907.86 soles, SD = 418.24 soles [equivalent of roughly 260 USD per month]). Most participants (54.5%) did not have a stable source of income; only nine (3.6%) reported that they lived "comfortably" or "well." Most women (67.7%) reported that they were not employed and not looking for work. On average, women were 20 weeks pregnant at their baseline interview ($M$ = 20.07, SD = 6.67, range: 4–37).

### Procedures

Participants were recruited into the study through social media, postings on the agency website and word-of-mouth. Participants reached out by phone to complete a brief screen with the project coordinator to determine eligibility and interest. The project coordinator provided a brief description of the study and intervention, and if interested, women were scheduled for a baseline interview. All interviews were conducted over the phone due to the pandemic, and thus, consent for participation was given verbally. For participants who were minors, parents gave verbal consent, and adolescents gave verbal assent for participation. Women completed four assessments in Spanish throughout the study, once at baseline (T1),

once following their participation in the intervention (T2), and at 3 and 12 months postpartum (T3 and T4, respectively). For each interview, participants received a small token of diapers and/or foodstuffs, valued at ~$10. Overall, study retention was excellent. Ninety percent of the sample completed the T2 interview, 82.0% completed the T3 interview and 77.7% completed the T4 interview.

Once women completed the baseline interview, they were assigned to receive one of two psychoeducational PH interventions (i.e., PH only [PH] content, or TF-PH content). Both interventions were delivered via WhatsApp over 5 weeks. For each text-based session, women received one set of brief psychoeducation materials, which included 150–350 words of text summarizing a set of brief psychoeducational videos (video durations ~5–7 min). The PH (control) condition included WhatsApp text-delivered content on nutrition, exercise during pregnancy, infant care, breastfeeding and mindfulness. Women in the TF-PH (intervention) condition additionally received WhatsApp text-delivered content about IPV against women, the effects of violence on mental health, safety planning and positive parenting. Treatment content for both conditions, including hyperlinks to all videos, can be found in Electronic Supplement 2. Both conditions were clinician-supported; that is, participants received the psychoeducational text messages through WhatsApp from a direct person-to-person text with a trained lay paraprofessional and could follow up with related questions. Lay paraprofessionals were staff members at the partner social service agency who had a history of work with women and families, but who were not health care professionals. They received ~10 h of training in perinatal mental health, IPV and related clinical supports. A psychologist was on-call for acute clinical referral needs. Regular team meetings occurred throughout the project to monitor progress and provide feedback to lay paraprofessionals on implementation.

### Measures

#### Demographics
Women reported on sociodemographic information, including age, income, employment, socioeconomic status and gestational age. Single motherhood was assessed by asking women, "Are you a single mother?"

#### Adverse childhood experiences
At baseline, women's experiences of childhood adversity, including both maltreatment and household dysfunction, were assessed using Felitti et al. (1998) 10-item ACEs Questionnaire. The ACEs Questionnaire has been previously used in Spanish, as well as in Peru (e.g., Carroll et al., 2021; Siego et al., 2021). Women reported whether each experience happened to them before the age of 18 years; responses were tallied so possible scores ranged from 0 to 10, where higher scores indicated exposure to more types of childhood adversity.

#### Intimate partner violence
At baseline, women's experiences of IPV were assessed using the *Escala de Violencia*, a scale developed in Mexico to measure physical, sexual and psychological IPV (Valdez-Santiago et al., 2006). While the scale has not been validated in Peru, it has been validated in Spanish, and aligns closely with how IPV is assessed in national demographic surveys in Peru (Ponce Gómez, 2012). Women responded to each of the 26 items using a 4-point Likert scale (0 = *Never*, 4 = *Many times)* to reflect their experiences in the past year. Internal reliability for the IPV assessment at baseline was $\alpha = .91$.

#### Post-traumatic stress symptoms
At all time points, PTSS was assessed using the *PTSD Checklist for DSM-5* (PCL-5; Weathers et al., 2013). The PCL-5's 20 items ask about symptoms of reexperiencing, avoidance, hyperarousal and negative mood and cognition, on a 5-point Likert scale (0 = *Not at all* to 4 = *Extremely*). Higher scores represent more severe symptoms. Scores above 32 indicate clinically significant symptoms. The PCL-5 has established evidence of psychometric validity in Spanish (Miller-Graff et al., 2022). Internal reliabilities for this study were $\alpha = .91, .88, .89$ and $.91$ (T1–T4, respectively).

#### Depression symptoms
Symptoms of depression were assessed at all time points using the *Center for Epidemiological Studies Depression* scale (CESD; Radloff, 1977). The CESD's 20 items ask about symptoms of depressed mood using a 4-point Likert scale (0 = *Rarely or none of the time*, 3 = *Most or almost all of the time*). Items were summed to create a total severity score with higher scores representing more severe symptoms; scores above 15 indicate clinically significant symptoms. The CESD has adequate psychometric validity in Spanish (Miller-Graff et al., 2022) and is a valid assessment of depression symptoms in the perinatal period (Heller et al., 2022). Internal reliabilities for this study were $\alpha = .87, .89, .88$ and $.88$ (T1–T4).

#### Multisystem resilience assets
Women's access to multisystem resilience assets was evaluated at all time points using the *Adult Resilience Measure* (ARM-R; Jefferies et al., 2018; Liebenberg and Moore, 2018). ARM-R items ask about access to or experiences of intrapersonal protective factors (e.g., personal skills), relational resilience (e.g., social support) and community/contextual factors (e.g., enjoyment of family and community traditions). For each of the 28 items, women responded using a 5-point Likert scale (0 = *Not at all like me* to 4 = *A lot like me).* Higher scores indicate higher resilience. The ARM-R has previously been used in Spanish (e.g., Clark et al., 2022). Internal reliabilities for this study were $\alpha = .90, .91, .88$ and $.87$ (T1–T4, respectively).

#### Parenting confidence
At all time points, parenting confidence was assessed using the *Karitane Parenting Confidence Scale* (KPCS). The KPCS assesses parents' confidence in their ability to parent effectively (Črnčec et al., 2008). Women responded to the 15 items on a 4-point scale (0 = *No, hardly ever* to 3 = *Yes, most of the time*). Higher scores reflect higher parenting confidence. A score below 40 indicates low parenting confidence that may warrant clinical support. Although the KPCS has been translated and validated in numerous languages, no Spanish-language version of the scale was available (Shrestha et al., 2016; Pereira et al., 2018). The scale was forward translated by a fluent Spanish speaker, reviewed by the local team, revised in conversation across the research-community partner team and finalized following a verification of semantic equivalence. Internal reliabilities for the current study were $\alpha = .77, .74, .76$ and $.72$ (T1–T4, respectively).

#### Treatment uptake
For each texted session in WhatsApp, lay paraprofessional facilitators indicated whether women had confirmed that they had received, accessed and reviewed the material. If participants confirmed this, the session was coded as completed (1), and if they did not respond or said that they had not reviewed the materials, the session was coded as not completed (0).

### Analytic plan

Given that randomization was not achieved, the current study leveraged PSM to match groups on key study variables to improve causal inference (Rubin, 1974; Schafer & Kang, 2008). PSM is a commonly used analytic method for causal inference in nonrandom designs that is used in a variety of disciplines (Caliendo and Kopeinig, 2008). PSM addresses the problem of selection bias in nonrandomized designs by creating a propensity score that "matches" subjects in the treatment and control groups and reduce the effects of confounders that may be associated with treatment (Caliendo and Kopeinig, 2008), resulting in the selection of a subsample of individuals within the treatment and control arms who have similar baseline characteristics. PSM approaches are preferred over more typical approaches using covariate-controlled regression when covariates are related to treatment status (Schafer & Kang, 2008).

In the current study, both of these conditions were met, and PSM was conducted using the PSMATCH2 package in Stata 17.0 (Leuven and Sianesi, 2018; StataCorp, 2021). PSM was modeled as a function of the month of participant enrollment relative to the start of the study and baseline values of ACEs and IPV exposure, income, status as a single mother and mental health (depression symptoms and PTSS) because local study staff indicated these were the factors they had considered when deciding whether to assign women to the PH-only or TF-PH condition. PTSS, depression symptoms, multisystem resilience assets and parenting confidence as outcomes. Nearest-neighbor matching (five closest) was used. Following PSM, the main effects of treatment at each time point were evaluated using multilevel modeling (time nested within persons) with random intercepts. Models were estimated with robust standard errors and propensity score weights applied at the person level. Treatment effects were evaluated using a modified version of Cohen's $d$ ($d_r$) developed for within-subject designs with random effects. Here, $d_r$ is calculated by dividing the unstandardized coefficients derived from the multilevel model by the SD of the person-level residual (Westphal, 2016). Post-hoc power analyses in Optimal Design suggested that the power reached .80 for an effect size of $d = 0.49$ if $n = 158$. Importantly, Optimal Design is designed to estimate power for randomized designs. As such, this power analysis is approximate rather than definitive, but indicates that the study is likely underpowered to detect small effects.

### Results

PSM resulted in the inclusion of $N = 150$ participants from the original sample of $N = 251$. Results of propensity score match diagnostics can be found in Electronic Supplement 3. Study flow for the matched sample can be found in Electronic Supplement 4.

### Adversity, violence and baseline mental health and parenting

The matched sample had high rates of exposure to adversity. Overall, 62.67% of women reported exposure to IPV in the past year, and 90.00% endorsed exposure to at least one ACE. Only 5.3% of the sample ($n = 8$) reported exposure to neither IPV nor any ACE. At baseline, 42.67% of women reported symptoms of depression that fell above the cutoff for clinical significance ($M = 15.50$, SD = 10.11). Only 6.67% of the sample endorsed PTSS that fell above the cutoff for clinically significant symptoms ($M = 13.53$, SD = 11.57). ACEs and IPV were both associated with depression ($r = 0.21$, $p = .007$; $r = 0.33$, $p < .001$), PTSS ($r = 0.34$, $p < .001$; $r = 0.33$, $p < .001$) and multisystem resilience ($r = -0.17$, $p = .034$; $r = -0.19$, $p = .022$). Neither ACEs nor IPV were significantly associated with parenting at baseline. Fifty-six percent of women reported levels of parenting confidence in the range considered sufficiently low to warrant clinical intervention ($M = 38.47$, SD = 3.68, range: 25–45). There are no standard clinical cutoffs for the assessment of multisystem resilience ($M = 106.14$, SD = 15.98).

### Treatment uptake

Participation in treatment was strong. On average, women participated in four of five sessions ($M = 4.30$, SD = 1.35). A few women did not participate in any sessions (4.38%), but most (77.69%) participated in all sessions. There was no significant difference between groups in the number of completed sessions ($t = 0.95$, $p = .341$) or in completion of each session (see Table 1).

### Differences in mental health, multisystem resilience and parenting across conditions

Across all models, there were no significant differences between conditions at baseline after the PSM (see Table 2). IPV was associated with higher depression symptoms ($B = 21.11$, $p < .001$), higher PTSS ($B = 22.77$, $p < .001$) and lower multisystem resilience ($B = -26.15$, $p = .001$). IPV was not significantly associated with parenting confidence. ACEs were not significantly associated with depression symptoms, PTSS, multisystem resilience or parenting confidence. Being a single mother was associated with higher depression symptoms ($B = 9.95$, $p < .001$), higher PTSS ($B = 10.02$, $p < .001$), lower multisystem resilience ($B = -10.24$, $p = .001$) and lower parenting confidence ($B = -2.56$, $p < .001$).

There were no significant differences between groups in depression, PTSS, parenting confidence or multisystem resilience over time. However, the current study is a pilot trial and the effects of perinatal psychoeducational interventions are likely to be small (e.g., $d = -0.31$ for mental health, Park et al., 2020). Given that the current study also did not include a no-treatment control, it is

**Table 1.** Treatment uptake data by session

| | Perinatal health | | Trauma-focused perinatal health | | |
|---|---|---|---|---|---|
| Session | Content | Attendance (%) | Content | Attendance (%) | Significant difference ($\chi^2$, $p$) |
| 1 | Nutrition | 96.10 | Nutrition and exercise | 95.89 | 0.004, $p = .947$ |
| 2 | Exercise | 94.81 | Violence and safety | 89.05 | 1.69, $p = .193$ |
| 3 | Infant care | 90.91 | Effects of violence, mindfulness and resilience | 83.56 | 1.83, $p = .176$ |
| 4 | Breastfeeding | 84.42 | Infant care | 76.71 | 1.43, $p = .232$ |
| 5 | Mindfulness | 74.03 | Breastfeeding and positive parenting | 73.97 | 0.0001, $p = .994$ |

**Table 2.** Multilevel models examining differences between prenatal health and prenatal health + trauma WhatsApp interventions

| | Depression | | | Posttraumatic stress | | | Multisystem resilience | | | Parenting confidence | | |
|---|---|---|---|---|---|---|---|---|---|---|---|---|
| | B (SE) | 95% CI | p | B (SE) | 95% CI | p | B (SE) | 95% CI | p | B (SE) | 95% CI | p |
| Group assignment | 1.05 (1.49) | [−1.86, 3.97] | .478 | −0.55 (2.08) | [−4.62, 3.53] | .792 | −0.34 (2.66) | [−5.54, 4.89] | .899 | 0.13 (0.57) | [−0.98, 1.24] | .821 |
| Time | | | | | | | | | | | | |
| Time 2 | −0.54 (1.19) | [−2.99, 1.80] | .650 | **5.36 (1.47)** | **[2.48, 8.24]** | **<.001** | 1.78 (2.02) | [−2.19, 5.75] | .379 | 1.74 (0.43) | [0.88, 2.58] | <.001 |
| Time 3 | 0.58 (1.51) | [−2.39. 3.54] | .702 | **5.05 (1.79)** | **[1.53, 8.57]** | **.005** | 0.64 (1.96) | [−3.20, 4.48] | .745 | 1.65 (0.52) | [0.63, 2.67] | .002 |
| Time 4 | 1.56 (1.65) | [−1.67, 4.79] | .343 | **8.56 (2.19)** | **[4.26, 12.84]** | **<.001** | 2.42 (2.03) | [−1.54, 6.40] | .231 | 1.80 (0.50) | [0.82, 2.78] | <.001 |
| Group assignment* time | | | | | | | | | | | | |
| Time 2 | 0.03 (1.62) | [−3.14, 3.21] | .983 | 1.16 (2.11) | [−2.99, 5.30] | .584 | 0.26 (2.46) | [−4.57, 5.09] | .916 | −0.26 (0.70) | [−1.64, 1.11] | .705 |
| Time 3 | −2.12 (2.19) | [−6.42, 2.18] | .335 | −0.79 (2.30) | [−5.49, 3.91] | .742 | 3.45 (2.51) | [−1.46, 8.36] | .168 | 0.35 (0.79) | [−1.19, 1.90] | .655 |
| Time 4 | −1.20 (2.21) | [−5.51, 3.13] | .587 | −1.42 (2.95) | [7.20, 4.36] | .631 | 1.77 (2.79) | [−3.70, 7.24] | .527 | −0.23 (0.78) | [−1.76, 1.30] | .770 |
| IPV | **21.11 (6.19)** | **[9.98, 33.24]** | **.001** | **22.77 (5.90)** | **[11.21, 34.33]** | **<.001** | **−26.15 (7.70)** | **[−41.25, −11.06]** | **.001** | −3.39 (1.79) | [−6.89, 0.12] | .058 |
| ACEs | 0.58 (0.33) | [−0.06, 1.23] | .076 | 0.83 (0.35) | [0.15, 1.51] | .016 | −0.71 (0.46) | [−1.60, 0.18] | .118 | −0.07 (0.09) | [−0.24, 0.11] | .463 |
| Single mom | **9.95 (1.99)** | **[6.05, 13.85]** | **<.001** | **10.02 (3.66)** | **[2.85, 17.18]** | **.006** | **−10.24 (3.17)** | **[−16.44, −4.03]** | **.001** | **−2.57 (0.71)** | **[−3.94, −1.18]** | **<.001** |
| Constant | 29.87 (4.01) | [22.02, 37.72] | <.001 | 28.11 (7.87) | [12.68, 43.54] | <.001 | 92.44 (6.54) | [79.62, 105.25] | <.001 | 34.23 (1.52) | [31.25, 37.21] | <.001 |
| *Random effect parameters* | Est. (SE) | 95% CI | | Est. (SE) | 95% CI | | Est. (SE) | 95% CI | | Est. (SE) | 95% CI | |
| SD (cons) | 5.68 (0.51) | [4.77, 6.76] | | 6.81 (0.90) | [5.26, 8.84] | | 9.36 (0.76) | [7.97, 10.98] | | 1.59 (0.20) | [1.24, 2.04] | |
| SD (residual) | 7.24 (0.38) | [6.52, 8.03] | | 9.24 (0.46) | [8.38, 10.20] | | 8.91 (0.38) | [8.20, 9.69] | | 3.00 (0.18) | [2.68, 3.37] | |

*Note:* Time 2 = ~6 weeks after baseline (post-treatment), Time 3 = 3 months postpartum, Time 4 = 12 months postpartum. Significant effects at *p*<.05 appear in bold.

**Table 3.** Treatment effect sizes

|  | Depression | Posttraumatic stress | Multisystem resilience | Parenting confidence |
|---|---|---|---|---|
| Time 2 | 0.004 | 0.125 | 0.029 | −0.087 |
| Time 3 | −0.292 | −0.085 | 0.387 | 0.117 |
| Time 4 | −0.166 | −0.152 | 0.198 | 0.073 |

*Note:* Time 2 = ~6 weeks after baseline (post-treatment), Time 3 = 3 months postpartum, Time 4 = 12 months postpartum.

reasonable to assume that effect size differences between groups would be small. As such, effect sizes were also calculated to provide further context (see Table 3). Overall, analyses suggested a slight advantage of the TF-PH condition, with effect sizes exceeding the cut-off for a small effect for group differences in women's depression symptoms ($d_r = -0.29$) and multisystem resilience ($d_r = 0.39$) at 3 months postpartum, both of which faded slightly at 1 year postpartum ($d_r = -0.17$ and $d_r = 0.20$, respectively). Effects of TF-PH on women's PTSS fell below the threshold for a small effect size but indicated possible slight elevations at post-test ($d_r = 0.13$) but possible improvements ($d_r = -0.15$) at 1 year postpartum, compared to the treatment group.

## Discussion

The current study aimed to evaluate the uptake and preliminary evidence for the additive benefit of trauma-focused content to a low-intensity digital psychoeducational intervention for pregnant women living in a resource-constrained context. The first objective was to describe uptake of the two intervention programs via engagement with WhatsApp-delivered treatment content. The second objective was to examine the differential effectiveness of the two programs on pregnant women's depression symptoms, PTSS, multisystem resilience and parenting confidence, controlling for exposure to IPV, ACEs and single motherhood.

Rates of exposure to violence for women in this context are nearly ubiquitous; 94.7% of participants reported exposure to childhood adversity and/or past year IPV, both of which were significantly associated with mental health and multisystem resilience at baseline. Women in the current study also reported high rates of depression and low levels of parenting confidence. This is consistent with the literature in both Peru and other global contexts (Gomez-Beloz et al., 2009; Malta et al., 2012). An important contribution of the current study, however, is the examination of the role of single motherhood – few previous studies in Peru or elsewhere have considered the experience of single motherhood on women's perinatal mental health. In the current study, single motherhood was associated with increased risk for mental health difficulties and low parenting confidence, as well as lower access to assets to bolster resilience. The experience of single motherhood varies significantly across contexts, but within Peru, there is a high level of stigma associated with single motherhood (Alvarado and del Carmen Vilchez, 2015), and Peruvian single mothers are exposed to high levels of poverty, unemployment and food insecurity (Santos et al., 2022). The lack of support for single mothers occurs across multiple systems, including the lack of available formal supports by the state and the lack of informal supports from their families. These findings emphasize the particular importance of providing supportive care to single mothers in Peru.

It should also be noted that the current study included pregnant women across a wide age span, including adolescents. Although the inclusion of adolescents is important, given the high rate of adolescent pregnancy in this context (Caira-Chuquineyra et al., 2024), adolescents may face unique developmental stressors associated with pregnancy, including poorer birth outcomes (Ventura et al., 2012) and stigma (Mori-Quispe et al., 2015). Although too few adolescents participated in the current study to analyze data separately, future research might consider a focus on pregnant adolescents in Lima to more fully articulate the needs of this population.

Overall, treatment uptake in the current study was excellent; 77.69% of participants participated in all five sessions. This is remarkably high compared to self-guided digital mental health interventions, which have been shown to have completion rates of 0.5–28.6% (Fleming et al., 2018). The current study contributes valuable information to the broader global literature on human-supported digital interventions, especially since reviews have noted that treatment uptake data have been infrequently and unreliably reported (Lipschitz et al., 2022).

Although the high levels of uptake are promising for the future of low-intensity interventions in this and similarly resource-constrained contexts, such interventions are unlikely to fully address women's mental health and parenting needs. There were no statistically significant differences between TF-PH and PH approaches in the current study, suggesting that the additional benefit of trauma-focused components in this format, if present, is small in magnitude. This is consistent with previous work on low-intensity perinatal mental health and digital interventions (Fu et al., 2020; Park et al., 2020). Since this was a pilot study and was thus underpowered to detect small effects, treatment effect sizes were examined (see Table 3). Overall, TF-PH had the strongest advantages in addressing maternal depression symptoms and resilience, with effect sizes in the small to moderate range at postpartum follow-ups. This finding mirrors another low-intensity digital mental health intervention in Latin America, which found that a brief, psychoeducational intervention delivered via WhatsApp yielded reduced depression symptoms over time among Brazilian older adults (Scazufca et al., 2024).

In the current study, there were negligible differences between conditions at immediate follow-up. Rather, the pattern of effects observed here suggests that effects were most robust at the 3-month postpartum follow-up and faded somewhat by 12 months postpartum. There is often variability across treatments in fade-out versus sustained versus sleeper effects (e.g., van Aar et al., 2017), underscoring the importance of long-term follow-ups in treatment research. The overall pattern of effects could possibly be explained by combined social and developmental contexts over the course of the study. That is, post-intervention follow-ups were conducted at the height of the coronavirus disease 2019 (COVID-19) pandemic and before women gave birth. By the 3-month postpartum follow-up, women had given birth, could have more actively implemented some of the learned skills and were perhaps more readily able to travel and access resources.

Results from the current study should be interpreted in light of a number of limitations. First, data were collected during the COVID-19 pandemic, which significantly impacted project operations, access to resources and numerous aspects of women's day-to-day lives and health care. The burden of disease from the pandemic was severe in Peru due to strict national lockdown measures, escalations in violence against women and a high rate of death and infection (Agüero, 2021; Herrera-Añazco et al., 2021; Cajachagua-Torres et al., 2022). Unfortunately, we do not have data on pandemic-related variations in maternal well-being due to the pandemic in this sample, and without replication, it is difficult

to determine the extent to which the pandemic may have impacted the findings of the current study. It is unknown whether intervention content overlapped with mental health services women may have received elsewhere, such as at PH care visits; it is therefore unknown if women received additional exposure to the intervention content from other sources during the study period. However, it is unlikely that many women received other mental health services during the study period due to the limited availability of such services in this district. Moreover, the lack of randomization, though mitigated through quasi-experimental approaches, reduced statistical power to examine differences between groups (since not all of the sample was "matched"). The implementation problem encountered regarding randomization, however, provides important insights for future research. The time and informality of in-person conversation likely allowed for broader discussions on project implementation and created space for this information to arise. It is rare for service-providing organizations to conduct random assignment to condition and underscores the necessity of shared conversations about the use of randomization. It also underscores the importance of robust (and in-person) training on research design, with intermittent monitoring checks on randomization, when randomization procedures are being carried out by lay paraprofessional staff. Several future directions for research and clinical work in this area are also evident. It would be helpful to gather data on more proximal outcomes (e.g., skills learned and used), which may have given better insight into the effectiveness of various treatment components. Ecological momentary analysis may be useful for the collection of such data (Singla, 2024). Measuring and analyzing other potential predictors of treatment uptake and effectiveness, such as obstetric history and PH outcomes, could be useful.

In sum, the current study described the results of a pilot trial of a digital intervention (i.e., via WhatsApp) implemented by local lay paraprofessionals to support perinatal mental health in an under-resourced community in Lima, Peru. Across both treatment conditions, intervention uptake was high, and findings suggest that past-year IPV and being a single mother were risk factors for participants' well-being. Although there were no significant differences between treatment groups over time, evaluation of effect sizes indicated that the treatment condition with PH content and added trauma-focused content (i.e., the TF-PH) yielded slightly better results in addressing participants' depression symptoms and multisystem resilience from baseline to 3 months postpartum. This study demonstrates that brief, digital interventions administered by lay paraprofessionals are feasible in this setting and may promote positive adjustment for maternal mental health and parenting in the perinatal period. This is particularly promising given the low availability of maternal mental health care in many global contexts and may support increased access to mental health supports during this key developmental juncture.

**Open peer review.** To view the open peer review materials for this article, please visit http://doi.org/10.1017/gmh.2025.10094.

**Supplementary material.** The supplementary material for this article can be found at http://doi.org/10.1017/gmh.2025.10094.

**Data availability statement.** Data are not open-access since participants did not consent to data sharing, but all study code and any meta-data required for the purposes of meta-analysis is available from the first author upon request.

**Acknowledgments.** The authors would like to thank the *Instituto de Pastoral de la Familia* for their collaboration and their commitment to the study participants. The authors would also like to extend their thanks to the participating women, who offered up their time during a particularly challenging period of their lives.

**Author contribution.** LMG was responsible for overall study design, procuring funding, conducting all analyses and leading the writing and revision of the manuscript. JC made significant contributions to project management, study design, data management and cleaning, as well as contributed to both the initial drafts and revisions of the manuscript. LYR was responsible for project oversight and coordination, contributed to study conceptualization and assisted with the writing and revision of the manuscript. EPC contributed to data collection and study implementation, data management and assisted with the writing and revision of the manuscript.

**Financial support.** The study was funded by the Kellogg Institute for International Studies and the Ford Family Program in Human Development Studies and Solidarity.

**Competing interests.** The authors declare none.

**Ethics statement.** Ethical approval for the current study was obtained from the University of Notre Dame IRB (protocol #19-04-5,333). All participants completed written consent.

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
