## [Reviewer Report]

I find this Project very important and timely. Peru is a difficult place for women, with glorification of motherhood and yet, lots of violence against women and children, harassment of women at all socioeconomic levels and for all shorts of reasons. We need tools like those designed by the authors. But:

1. The age of participants ranges from 16 to 44. This puts adolescent mothers in the same group as adults, and they are different because adolescents face development stressors,at the same time as stigma and a higher probability that their pregnancies have resulted from sexual abuse.

2. The escala de violencia has not been validated in Peru

3. Very surprising that adverse childhood effects are not related to depression, Post traumatic stress symptoms and resilience. I wonder if this has to to do with the inclusion of teenagers in the same sample.

4. Single motherhood- how is it operationally defined? Not currently living with a partner? If so, did they ever cohabit and he left? Or they never lived together because, for instance he refused to accept the pregnancy as his? Would this include women living with a partner who is not the biological parent of the index pregnancy? – again including teenager pregnant women in the sample may distort the results.

5. The slightly more positive effects in the TF- PH group seemed to fade out. In what way was this related to the context of the pandemic (the longest lockup including school closure, the highest rate of excess death in that period, very stressful media treatment of what was happening).

---

## [Reviewer Report]

1. Please clarify why 14 years old was chosen as the minimum age criterion for participant inclusion.

2. Specify the ethical procedures followed for including minors in the study, such as obtaining parental or guardian consent and participant assent.

3. Clarify the professional background of the lay paraprofessionals who delivered the interventions. Indicate whether they were healthcare professionals or non-professionals, and explain if their performance was evaluated.

4. Line 36 (rather than just a subset of those at more acute risk; Tanner-Smith et al., 2018), Check citing.

5. About the intervention, this should be described clearly in one part (ie, clarifying if the session was a WhatsApp content message).

6. Indicate whether information about mental health services received by pregnant women during their routine health center check-ups was recorded, and clarify if these services overlapped with the study intervention period.

7.I suggest including a diagram or flowchart to illustrate the data collection process, as the current description is difficult to follow due to changes made during the study.

8. In all tables, please define the abbreviations T2, T3, and T4 for clarity.

---

## [Reviewer Report]

1. “There are only two state hospitals in San Juan de Lurigancho, prompting difficulties in accessing healthcare, such as prenatal or labor and delivery care (Perleche-Ugás et al., 2022).” page 11. It would be beneficial if this information were complemented by the number of health centers and maternal health centers. Typically, accessing hospitals requires a referral, particularly for labor and delivery care. Even though community service exists and the number of centers might have increased, sometimes access is limited (poverty, location, lack of resources, and professionals).

2. Page 18. “Fifty-sis percent” - Fifty-six percent

3. Page 22- 23 Check rewriting one sentence to increase clarity. “Given the limited availability….

4. While the word ‘doses’ is understandable, in this context it may be better to use ‘sessions’ or another term that more accurately reflects the intended meaning in this paragraph

---

## [Reviewer Report]

There are no further comments for changes. I believe the article demonstrates the effort to improve mental health care in vulnerable areas and also shows an initiative to promote innovation within a complex healthcare system.